# *Escherichia coli* cells evade inducible parE toxin expression by reducing plasmid copy number

**Shengfeng Ruan,[1] Christina R. Bourne[1]**

**ABSTRACT** Plasmids play important roles in microbial ecosystems, serving as carriers of antibiotic resistance and virulence. In the laboratory, they are essential tools for genetic manipulation and recombinant protein expression. We uncovered an intriguing survival phenotype in a fraction of the bacterial population while using plasmid-mediated arabinose-inducible gene expression to monitor the production of toxic ParE proteins. This phenotype was not correlated with changes to the plasmid sequence and could not be rescued by increasing arabinose uptake. Instead, survival correlates with a marked reduction in plasmid copy number (PCN). Reduced PCN is reproducible, not a function of the pre-existing population, and can be sequentially enriched by continual passage with induction. The reduction in PCN appears to allow mitigation of toxicity from the expression of ParE proteins while balancing the need to maintain a threshold PCN to withstand selection conditions. This indicates an adaptive cellular response to stressful conditions, likely by altering the regulation of plasmid replication. Furthermore, this survival mechanism appears to not be limited to a specific bacterial strain of *Escherichia coli* or ParE toxin family member, suggesting a generalized response. Finally, bacterial whole genome sequencing indicated an N845S residue substitution in DNA polymerase I, which correlates with the observed reduction in PCN and has been previously reported to impact plasmid replication. Further understanding this molecular mechanism has broader implications for this adaptive response of the dynamics of plasmid-mediated gene expression, microbial adaptation, and genetic engineering methodologies.

**IMPORTANCE** This research has increased our understanding of how bacteria respond to the pressure from plasmid-borne toxic genes, such as those found in toxin-antitoxin systems. Surprisingly, we found that bacteria survived toxic ParE protein expression by reducing the number of these plasmids in the cells. This discovery reveals another way in which bacteria can balance toxin expression with antibiotic selection to attenuate the effects of deleterious genes. This insight is not only valuable for understanding bacterial survival strategies but may also influence the development of better tools in biotechnology, where plasmids are often used to study the functional roles of genes.

**KEYWORDS** ParE toxin, plasmid copy number, origin of replication, recombinant expression, arabinose induction

P lasmids are circular, double-stranded DNA molecules that are naturally present in the microbial world, including bacteria and archaea (1). They exist as a physically distinct entity from the cell's chromosomal DNA that can replicate independently and have evolved over time as part of these organisms' genetic makeup. These natural plasmids often bring their hosts beneficial traits, such as antibiotic resistance and virulence, thereby offering survival advantages in competitive environments (2–5). They are also important drivers of bacterial evolution due to their intercellular transfer by conjugation or mobilization (6–8).

Address correspondence to Christina R. Bourne, cbourne@ou.edu.

The authors declare no conflict of interest.

See the funding table on p. 17.

In the laboratory, plasmids are important tools in genetic engineering procedures such as gene cloning and recombinant protein expression. Laboratory-based plasmids canonically possess a minimal set of essential components, including an origin of replication (*ori*), a selection marker, and a cloning site for the insertion of genes of interest. In order to control the expression of an inserted recombinant gene, these recombinant plasmids frequently incorporate catabolic repression systems as inducible promoters (9). The arabinose-inducible (*ara*BAD) promoter is commonly used and is tightly controlled by the presence or absence of arabinose in the environment (10). Notably, due to catabolite repression, the induction of *ara*BAD is easily repressed by the presence of excess glucose or lactose (11). This specificity ensures that the induction of the *ara*BAD promoter is tightly controlled and limited to the presence of arabinose, enabling modest induction levels for toxic protein expression (12), making it more suitable for certain applications.

The ease of modifying recombinant plasmids (13, 14) as well as their capacity for self-replication and tunable recombinant protein expression (15, 16) within host cells render them useful for ectopically expressing recombinant proteins in bacteria. Although, under particular conditions, the host cells benefit from the survival advantages conferred by the plasmid, the presence of plasmids can impose a metabolic burden on the host cells, potentially leading to adverse effects on plasmid stability and quality (17, 18). Given the important role of plasmid stability in ensuring research reliability and validity, the problem of plasmid instability is a concern. This is especially the case for studies expressing toxic proteins, such as those from toxin-antitoxin (TA) systems, where standard characterization sometimes relies on demonstration of toxicity upon induction of toxin and rescue upon induction of antitoxin (19–22). We have previously noted toxins that are active in their native hosts but were not toxic in the commonly used *Escherichia coli* surrogate host (22), and in the current work, we have uncovered additional confounding issues with these strategies.

Our research focuses on the ParE toxin proteins, components of the type-II ParDE TA system. It is one of the few known toxin proteins that target DNA gyrase (23), an essential enzyme involved in DNA replication and maintenance (24–26). The first ParDE system to be characterized is carried on the broad host range low copy number RK2 plasmid (27). In this capacity, the TA system serves an addiction function to enforce maintenance of the plasmid through the gyrase-inhibiting function of the ParE toxin. However, it was reported that under some growth conditions, the plasmid could be lost, and, paradoxically, viability was restored over time. Later, other ParE family members were studied, including one encoded within a pathogenicity island on the chromosome of *Mycobacterium tuberculosis* (Mt) (28). It was found that the overexpression in an *E. coli* host caused potent toxicity but was followed by a full recovery of growth.

Our studies have also observed a robust reduction in cell viability upon ParE protein expression but, in some instances, this is followed by a recovery of viable cell counts, indicating some survival mechanism may be induced. To investigate the underlying mechanism of this recovery phenotype, ParE proteins were cloned from the Mt chromosome (Mt ParE1 and Mt ParE2) and were ectopically expressed in *E. coli*. These exerted maximum toxicity on cells within 4 h of induction, after which cultures either maintain a steady population or recover growth, which is consistent with previous observations (28). Unexpectedly, we observed an insensitivity to subsequent re-induction of ParE protein expression after initial exposure and recovery. Through culture-based assays, we find that the loss of toxicity upon re-induction of ParE protein expression does not result from mutations of the plasmids or by alteration of arabinose uptake. Instead, we captured cells within the population with a consistent and stably reduced plasmid copy number (PCN); furthermore, this phenotype could be enriched from the culture population with subsequent ParE exposure via continual induction. Our investigation suggests that the observed reduction in PCN may be attributed to a mutation at the N845S residue in DNA polymerase I and potentially the ratio of replication regulators

RNA I and RNA II. These factors are intricately connected to the replication process of the plasmid under investigation.

In summary, our studies highlight a pervasive reduction in PCN, which could be a confounding variable for standard studies of recombinant toxic proteins in engineered plasmids. We have thus characterized a phenotypic balance between two toxic contrasting outcomes: maintaining PCN to support survival by ensuring sufficient expression of the antibiotic-resistant selection maker versus reducing the PCN to limit the expression of the ParE toxin protein.

## RESULTS

### A subset of cells within a population exhibit survival during ectopic ParE1 protein expression

The expression of the ParE1 toxin protein, derived from *Mycobacterium tuberculosis*, was induced in *Escherichia coli* MG1655 cells at multiple induction strengths (Fig. 1A). Induction by arabinose relied on an inserted arabinose-inducible (*ara*BAD) promoter, cloned from a pBAD vector to replace the existing *tetRO* promoter on the pMind vector, generating the "pMindBAD" plasmid. The resulting colony-forming units per milliliter (CFUs/mL) sampled over time revealed a dose-dependent decline in cell viability, confirming the toxicity of ParE1 protein expression to cells (Fig. 1B). Under a standard liquid growth condition in Luria-Bertani broth, a 0.02% induction level led to a 1.3-log reduction in cell counts within 4 h, while a 2% induction resulted in a more pronounced 3.1-log reduction in the same time frame (Fig. 1B). In contrast, when no arabinose was added during the same time frame, cell counts increased by 1.4-fold, which aligns with the 1.6-fold increase in cell counts for *E. coli* cells harboring the pMind "empty" vector lacking the inserted ParE1-encoding gene over the same 4-h period (Fig. S1). However, the induction of ParE1 protein expression did not result in a complete cell loss; instead, cells demonstrated the ability to either recover growth or maintain a steady population beyond 4 h post-induction (Fig. 1B).

To further investigate the underlying mechanism of this phenotype, following the 24-h induction period, one colony from each induction strength (designated as surviving cell "S" and corresponding arabinose concentration: S0, S002, S02, and S2, Fig. 1A and C through F) was isolated and re-grown to the stationary phase in the absence of the arabinose inducer. Subsequently, the stationary culture was diluted 1:20, and ParE1 protein expression was re-induced with the same inducer concentrations (Fig. 1A). Unexpectedly, the re-induction of ParE1 protein expression exerted markedly two different toxicity profiles: cells originating from 0% induction remained sensitive to the induction of ParE1 protein expression (S0, Fig. 1C) as did the culture originating from the 0.2% induced surviving colony (S02, Fig. 1E). However, there was no sensitivity remaining for the cells originating from 0.02% induction (S002, Fig. 1D) and 2% induction (S2, Fig. 1F) conditions. This unexpected differential response to ParE protein re-induction implies that the recovery in cell growth was not solely a consequence of the depletion of arabinose inducer at longer growth time points. It emphasizes the complexity of the cellular response to ParE1 toxin expression and prompts further exploration into the underlying mechanisms.

### Loss of re-induction sensitivity does not result from plasmid mutations

As a gyrase inhibitor, the action of ParE toxins raises the possibility of mutation-prone repair as a means of cell rescue and subsequent survival (29). Furthermore, previous studies have indicated mutations in promoter regions as a common means for cellular escape of toxin effects (30). To investigate if mutations on the plasmid contributed to the observed diminished toxicity profile, plasmids from the surviving cells S002 and S2 (designated as plasmids p002 and p2) were extracted and purified. These purified plasmids were subsequently transformed into the "starting" *E. coli* MG1655 cells, and viability was assessed over time with 0% or 2% arabinose (Fig. 1A). Surprisingly, the

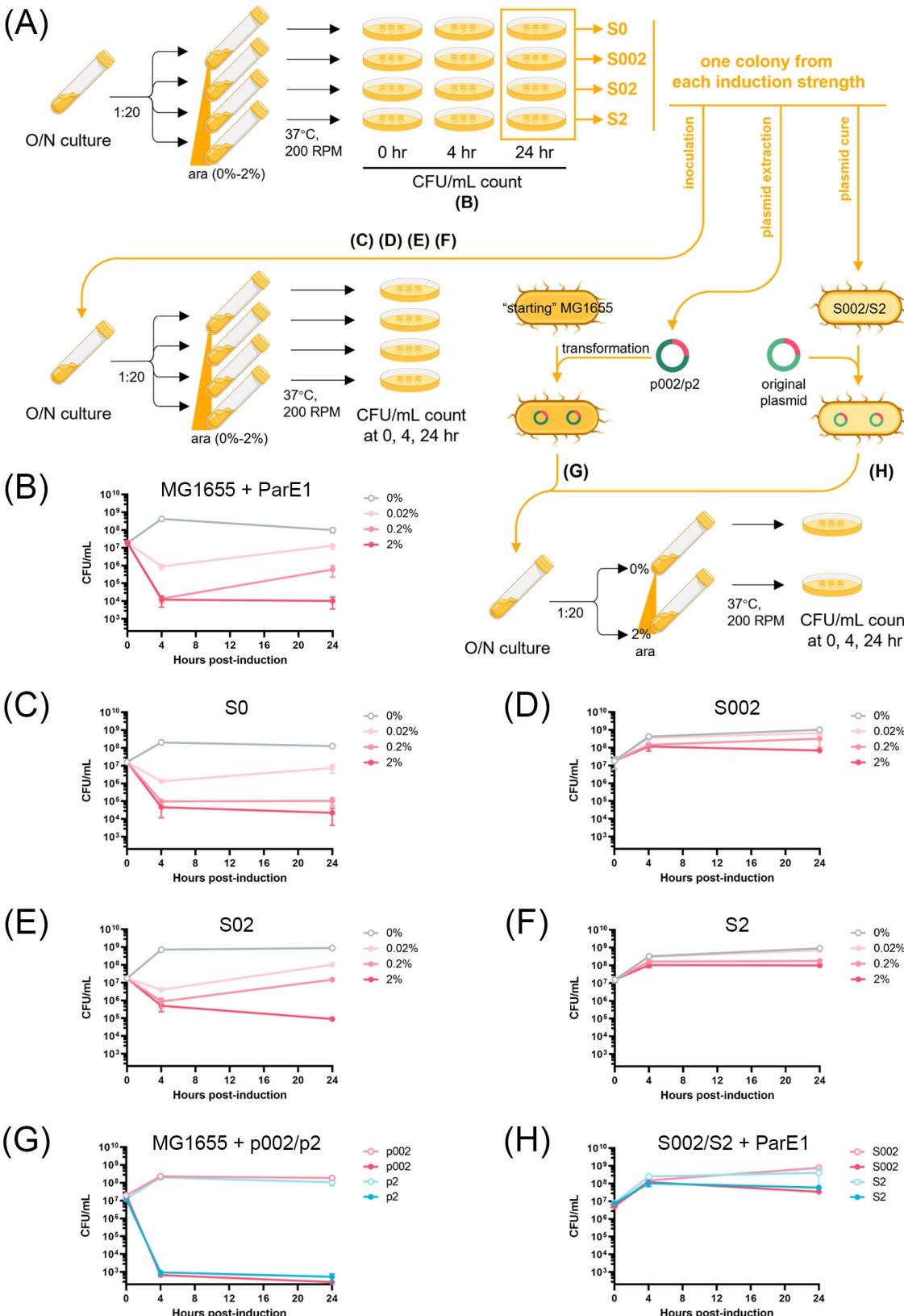

**FIG 1** Impact of the induction of ParE1 protein expression on *E. coli* cell viability. (A) Schematic representation of the experimental workflow. ara, arabinose. (B) The induction of ParE1 protein expression in *E. coli* MG1655 cells exhibited an induction-strength-dependent pattern. (C–F) Following a 24-h induction of (Continued on next page)

**FIG 1** (Continued)

ParE1 protein expression, the reinduction of ParE1 protein expression in the isolated cells exerted two different toxicity profiles: cells S0 (C) and S02 (E) remained sensitive to the reinduction of ParE1 protein expression, while cells S002 (D) and S2 (F) lost the sensitivity. (G) The induction of ParE1 protein expression from the extracted plasmids exhibited a strong toxicity profile to the "starting" *E. coli* MG1655 cells. (H) Plasmid-cured S002 and S2 cells lost the sensitivity to the induction of ParE1 protein expression from the transformed original plasmid construct. Each data point is presented with the standard error of the mean, representing at least two independent experiments.

growth profiles of both S002 and S2 cells exhibited similar sensitivity to the re-induction of ParE protein expression (Fig. 1G) as the original ParE1-expressing cells (Fig. 1B), suggesting the plasmid was unlikely to contain mutations that would prevent ParE1 protein expression.

In parallel, S002 and S2 cells were cured of the plasmids via sequential plating without selection pressure. After confirming the plasmid removal through the absence of quantitative polymerase chain reaction (qPCR) amplification and restored sensitivity to the selective antibiotic, these surviving cells were re-transformed with the original ParE1-expressing plasmid construct, and viability was assessed over time with 0% or 2% arabinose (Fig. 1A). As shown in Fig. 1H, these cells did not regain the sensitivity to the induction of ParE1 protein expression, indicating the phenotype is associated with the *E. coli* culture rather than with the plasmid.

Attempts to directly sequence the plasmids p002 and p2 were not successful, likely due to a noted reduction in yields. Therefore, polymerase chain reaction was used to amplify the full plasmid using the extracted samples as templates, and two different priming regions were used to ensure overlapping full coverage (Table S2; Fig. S2. pMind + araC_FWD with araC-pBAD_REV and araC-pBAD_FWD with pMind + araC_REV). Nanopore sequencing revealed no alterations in the base sequences as compared to the starting sample, further confirming that the lack of sensitivity to re-induction was not associated with plasmid defects.

These findings collectively indicated that the observed difference in toxicity profiles between the surviving cells and their original counterparts is not attributed to mutations within the plasmids. Instead, other factors, possibly related to cellular adaptations or responses, including plasmid maintenance mechanisms, could be influencing the reduced impact of ParE1 protein expression on viability. This speculation shifts the focus from genetic changes in the plasmid to possibly epigenetic or physiological changes within the cells that confer a survival advantage induced by the presence of the ParE1 toxin protein.

## The phenotypic loss to induction sensitivity cannot be complemented by increasing inducer uptake

In the recombinant pMindBAD plasmid system, expression of the ParE1 protein is contingent on the presence of arabinose inducer. This system is sensitive to the intracellular levels of arabinose, which are regulated by specific transport proteins, primarily the AraE permease (31). To evaluate the arabinose uptake and its subsequent effect on protein expression, a reporter pHerd20T plasmid encoding the fluorescent protein mCherry downstream of the same *ara*BAD promoter was introduced by transformation into the plasmid-cured surviving cells S002 and S2, as well as into the "starting" MG1655 *E. coli* cells. Induction of mCherry protein expression was achieved through the addition of 2% arabinose, followed by visualization using fluorescence via microscopy. As expected, the MG1655 cells that had not previously been subjected to ParE1 protein expression exhibited clear fluorescence within cells across the observed field and this was dependent on the inducer (Fig. 2A), indicating efficient mCherry expression. In contrast, the fluorescence was notably absent in the ParE1 plasmid-cured S002 and S2 cells, even with the highest level (2%) of arabinose inducer (Fig. 2A), suggesting a marked reduction in mCherry expression within these cells.

To quantify this observation at the mRNA level, subsequent reverse transcription-quantitative polymerase chain reaction (RT-qPCR) assays were conducted to measure

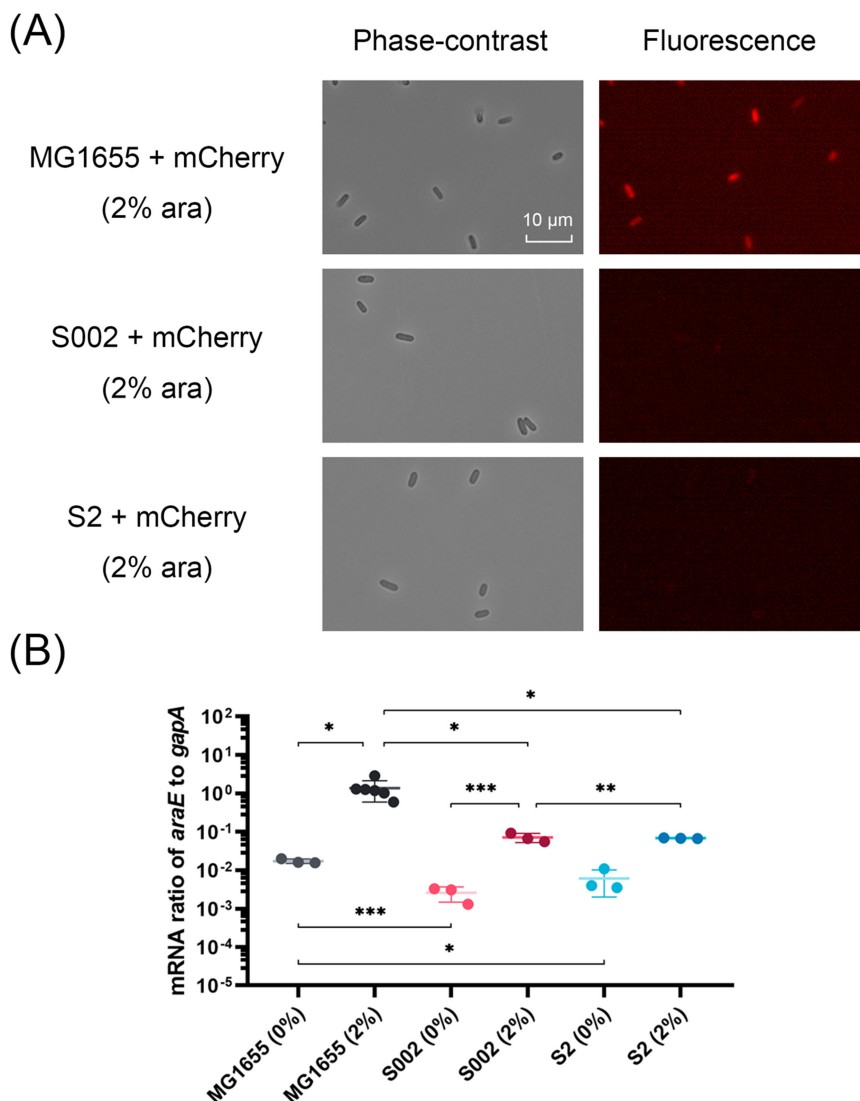

**FIG 2** *araE* mRNA levels are sensitive to arabinose induction in ParE1-surviving cultures, despite the lack of fluorescent protein mCherry signal. (A) Fluorescence microscopy of the "starting" MG1655 as well as the S002 and S2 cells with ParE-expressing plasmid removed ("cured") induced for the expression of fluorescent protein mCherry from a plasmid, pHerd20T, also under the control of the arabinose-inducible promoter. However, the fluorescence of mCherry protein is only visualized in the "starting" MG1655 cells. ara, arabinose. (B) The mRNA ratio of the arabinose transporter *araE* to chromosomal *gapA* in these cells after incubation with 0% and 2% arabinose. Each measurement with standard deviation contains at least three technical replicates. Unpaired two-tailed Student's *t*-test was performed. *$P < 0.05$; **$P < 0.01$; and ***$P < 0.001$.

*araE* mRNA levels in cells. This unveiled a consistent pattern of upregulated *araE* transcription level following the addition of arabinose in both "starting" and surviving cells, with an approximate 80-fold increase in MG1655 cells, an approximate 28-fold increase in S002 cells, and an approximate 11-fold increase in S2 cells (Fig. 2B). However, despite this consistent pattern of transcriptional upregulation, the *araE* mRNA levels were significantly lower in the surviving cells than in the "starting" cells both before and after upregulation. To further investigate if these decreased transcription levels affect mCherry expression, a secondary pRK2 plasmid constitutively expressing AraE protein was co-transformed into these surviving cells. However, although the presence of the AraE-expressing plasmid significantly increased the transcription level of *araE*, the

fluorescence was still absent in these surviving cells (Fig. S4). Despite this transcription increase, the absence of fluorescence in the surviving cells implies a discrepancy between *araE* mRNA levels and functional mCherry protein expression. When integrated with the observation of inefficient mCherry expression in these surviving cells, this insight underscores that arabinose transport remains partially functional despite the alterations introduced by ParE1 toxin expression.

## Cells modulate plasmid copy number to survive ParE1 protein expression

Persistently low yields when extracting plasmids from these surviving cells suggested that the plasmid copy number may have undergone alterations. To investigate this, the change of PCN in cells before and after ParE1 protein exposure was determined using quantitative polymerase chain reaction analysis. The ratio of plasmid to chromosome (P/C) was calculated using efficiencies of the individual primers and the threshold cycle (Ct) values arising from the *parE1* gene on plasmids and the chromosomal *gapA* gene. Two different primer sets, targeting the plasmid *ori* region or toxin gene (Table S2; Fig. S2; ori-qPCR and parE1-qPCR primer sets), were used. The data indicated a consistent result irrespective of which plasmid region was detected (Fig. S5). Strikingly, when compared to the pre-induction MG1655 cells, the surviving cells S002 and S2 exhibited an approximate 40-fold reduction in PCN, resulting in an average change from 697.4 ± 163.4 plasmid copies per chromosome pre-induction to 17.3 ± 9.4 plasmid copies per chromosome with re-induction insensitivity (Fig. 3). It suggests that the reduction in PCN can be a survival mechanism, potentially diminishing the ParE1 toxin's impact by reducing the number of toxin protein molecules produced per cell.

## There is a direct correlation between reduced plasmid copy number and survival phenotype

The survival phenotype was limited within the population of the "starting" cells, with only two out of the four tested colonies (i.e., S002 and S2 cells) persisting during re-induction (Fig. 1C through F). To assess if this persisting fraction could be further enriched, following the initial 24-h induction, cultures of MG1655 *E. coli* cells expressing ParE1 toxin were subjected to successive passages in fresh medium while maintaining the arabinose inducer at the same concentration (Fig. 4A). Notably, in the three groups subjected to induction with arabinose, the cultures displayed progressively higher cell viability with each successive passage at all induction strengths (0.02%–2%) (Fig. 4B). This suggests that the population of induction-surviving cells becomes more predominant over successive passages.

Following the second passage, two colonies from each induction strength were isolated and re-grown in the absence of the arabinose inducer. Cultures were diluted at a 1:20 ratio, and ParE1 toxin expression was subsequently re-induced with 2% arabinose. Strikingly, upon re-induction, the toxicity profiles of all re-grown colonies with previous ParE1 toxin exposure remained viable at all induction concentrations (0.02%–2%) (Fig. 4C). In contrast, the colonies derived from cells with no prior ParE1 exposure remained as sensitive as the initial observation, with only a minor fraction surviving (Fig. 4C, colonies S0%).

Quantification of PCN for the cultures of these surviving colonies shows a consistent reduction for all surviving cells (Fig. 4D), with a more pronounced reduction correlated with higher inducer concentration. For cells maintained at a 0.02% inducer concentration, there was an approximate 53-fold reduction, resulting in an average from 697.4 ± 163.4 plasmid copies per chromosome for sensitive cells to 13.1 ± 2.7 plasmid copies per chromosome in insensitive cells. For the cells maintained at 0.2% inducer concentration, there was an approximate 50-fold reduction, resulting in an average of 11.3 ± 2.1 plasmid copies per chromosome. However, for cells maintained at 2% inducer concentration, the PCN was below the limit of quantification (a P/C ratio of 0.25, corresponding to one plasmid to four chromosomes) even under constant antibiotic selection pressure. These data collectively provide evidence that the strategy of survival through the reduction of

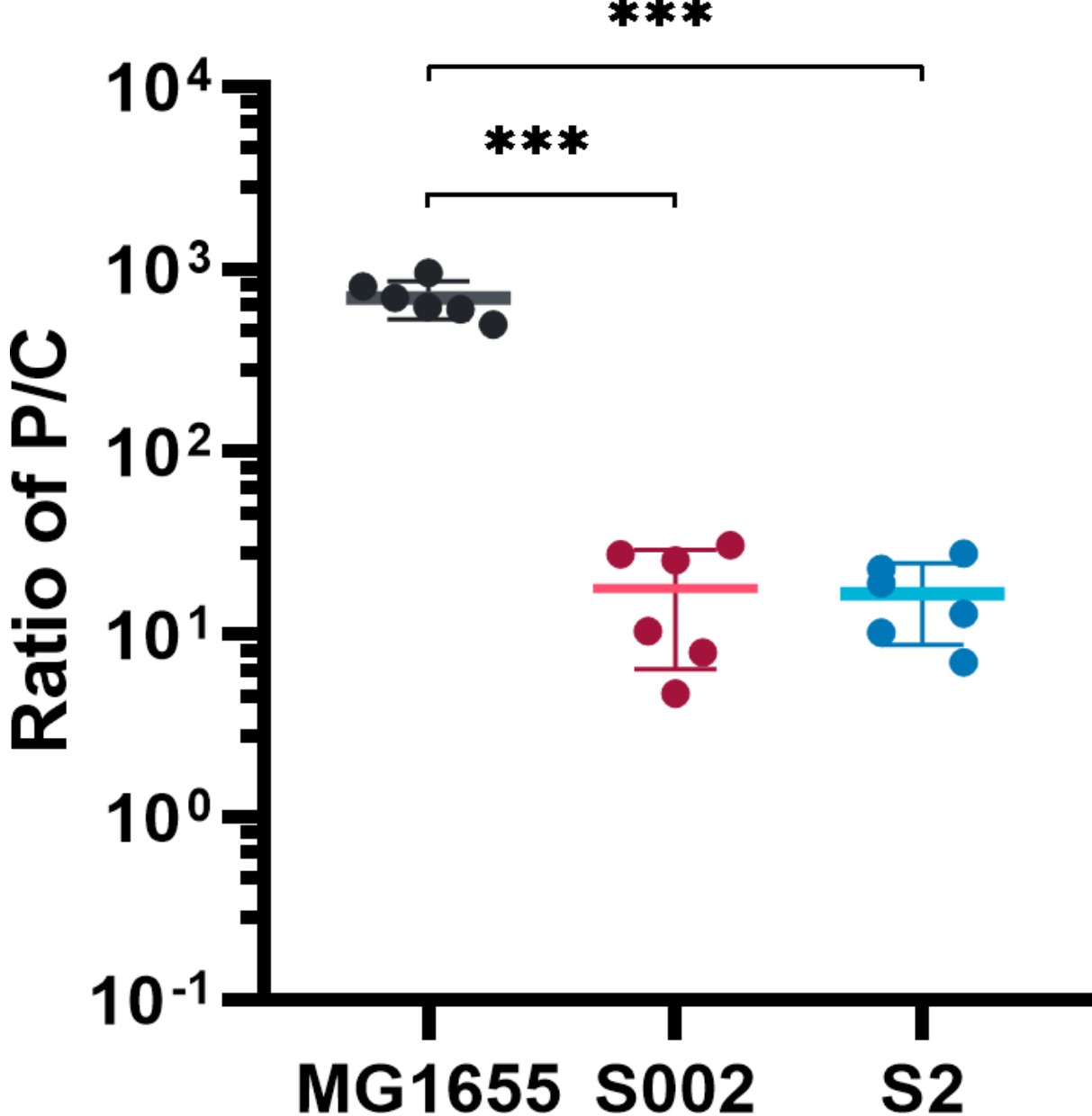

**FIG 3**  Cells modulate plasmid copy number to survive ParE1 protein expression. The ratio of P/C was defined as the amplification signal ratio of plasmid to chromosome, as described in Materials and Methods. The surviving S002 and S2 cells exhibited a significantly reduced PCN, compared to the pre-induction MG1655 cells. Each measurement with SD represents two independent experiments, with each containing three technical replicates. Unpaired two-tailed Student's *t*-test was performed. ***$P < 0.001$.

PCN to mitigate toxin expression is a reproducible phenomenon directly correlated with the amount of ParE1 protein induction.

We questioned if toxicity was the driver for reduced PCN. Experiments were carried out using a control pMind plasmid without the insertion of the ParE1-encoding gene. When this "empty" plasmid was introduced into the "starting" MG1655 *E. coli* cells, no reduction in PCN was found upon arabinose induction (Fig. S6A). Considering the inconsistency between the promoters on this control plasmid and the pMindBAD plasmid, another control pHerd20T plasmid was introduced, which contains the same *ori* and *ara*BAD promoter but with a non-toxic fluorescent protein mCherry gene. Consistently, no reduction in PCN was found (Fig. S6B). This demonstrated that the decrease in

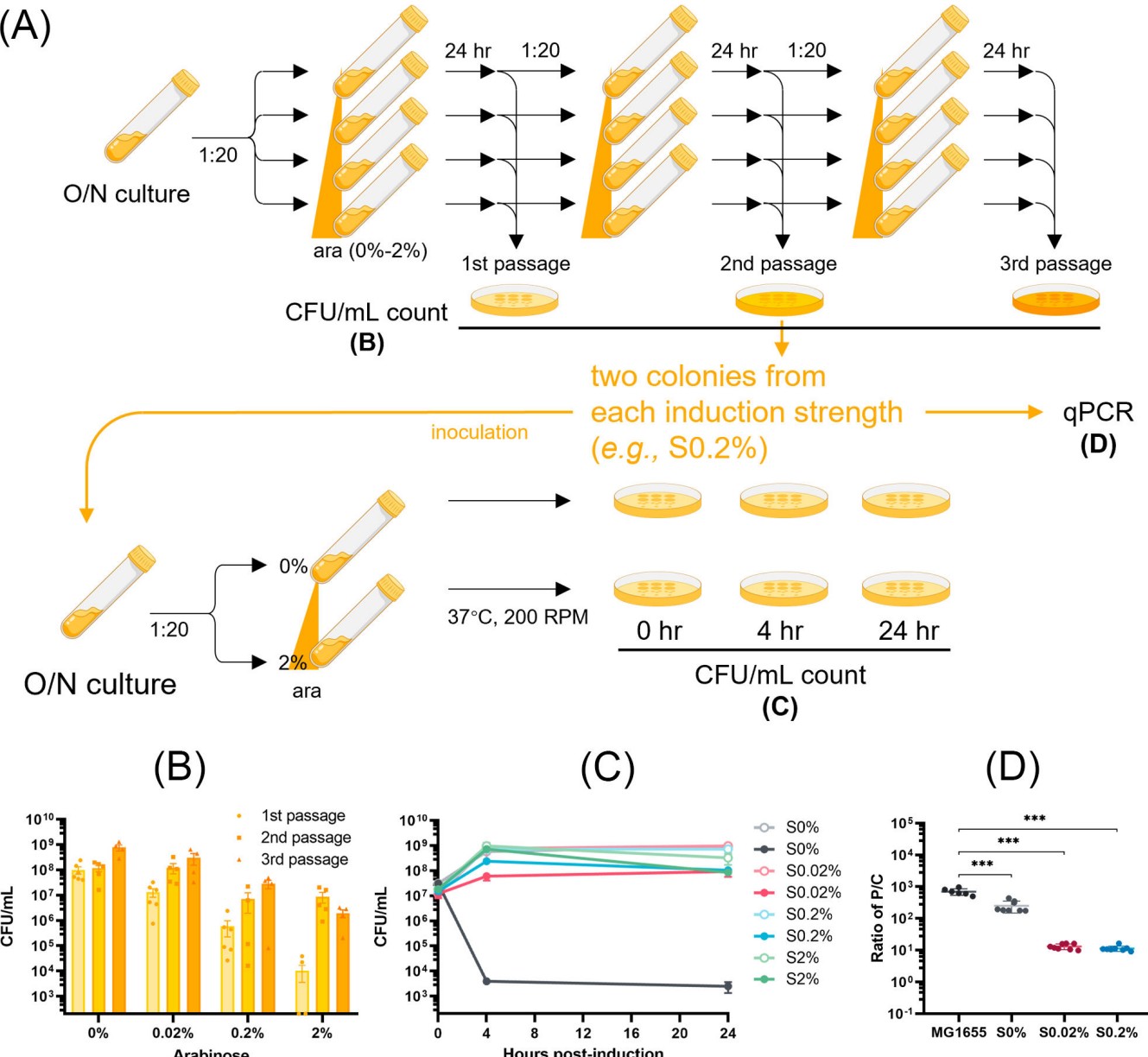

**FIG 4** Assessment of *E. coli* MG1655 cell viability and plasmid copy numbers upon induction of ParE1 toxin expression. (A) Schematic presentation of the passage workflow. ara, arabinose. (B) Cultures of MG1655 cells harboring ParE1-expressing plasmids were passaged every 24 h at the indicated induction levels, and all successive cultures had increased numbers of viable cells. (C) Cell viability was subsequently determined for two colonies isolated after two passages and re-induced with the addition of 0% (hollow circle) or 2% (solid circle) arabinose. All re-grown colonies apparently exhibited the same growth pattern at 0% reinduction, indicating there were no growth defects for those nonsensitive cells. Each data point represents the mean value of the two colonies, shown with SEM. (D) The PCNs of these surviving cells were determined by qPCR analysis. All cells that lost partial or complete sensitivity to reinduction exhibited a decreased PCN. Graph shows SD from two independent experiments performed in triplicates. Unpaired two-tailed Student's *t*-test was performed. ***$P < 0.001$.

PCN, and hence cell survival, was specifically driven by the toxic expression of ParE1 protein and not merely by the presence of the plasmid or the induction process. Collectively, these results indicate that survival of ParE1 protein-expressing cells is driven by reducing PCN, thereby limiting the available DNA template for transcription.

## Reduced plasmid copy number is a generalized phenotype for MG1655 *E. coli* survival of ParE protein expression

To further explore whether this phenomenon is limited to only the specific ParE1 toxin protein, the ParE2 toxin protein from Mt was cloned into the same pMindBAD plasmid (Fig. S3). While these two toxin proteins belong to the same family and both exert their impacts by the inhibition of DNA gyrase, their protein sequence similarity is limited to 37% (Fig. S7). The viability of MG1655 cells expressing ParE2 protein was monitored following induction with arabinose, again documenting a toxicity profile that correlated to induction strength and, notably, displayed even greater potency than ParE1 protein. At 2% induction, the ParE2 protein expression induced a complete cell loss at 24 h post-induction, while at 0% induction, presumably leaky expression of ParE2 protein resulted in a lower CFU when compared to ParE1 protein (Fig. 5A). As observed with ParE1 protein, cultures slowly recovered growth between the 4- and 24-h time points.

Given the stronger toxicity of the ParE2 protein, a lower concentration range of arabinose was employed for subsequent culture passages. As shown in Fig. 5B, the culture CFU counts have a pattern similar to that observed with ParE1 protein, albeit at lower induction strengths. However, the population of cells exposed to the highest induction (0.02%) did not exhibit a progressive enrichment, and instead, essentially the same number of CFU counts was obtained for each passage. This could potentially be attributed to the heightened toxicity of ParE2 relative to ParE1 at the 0.02% induction

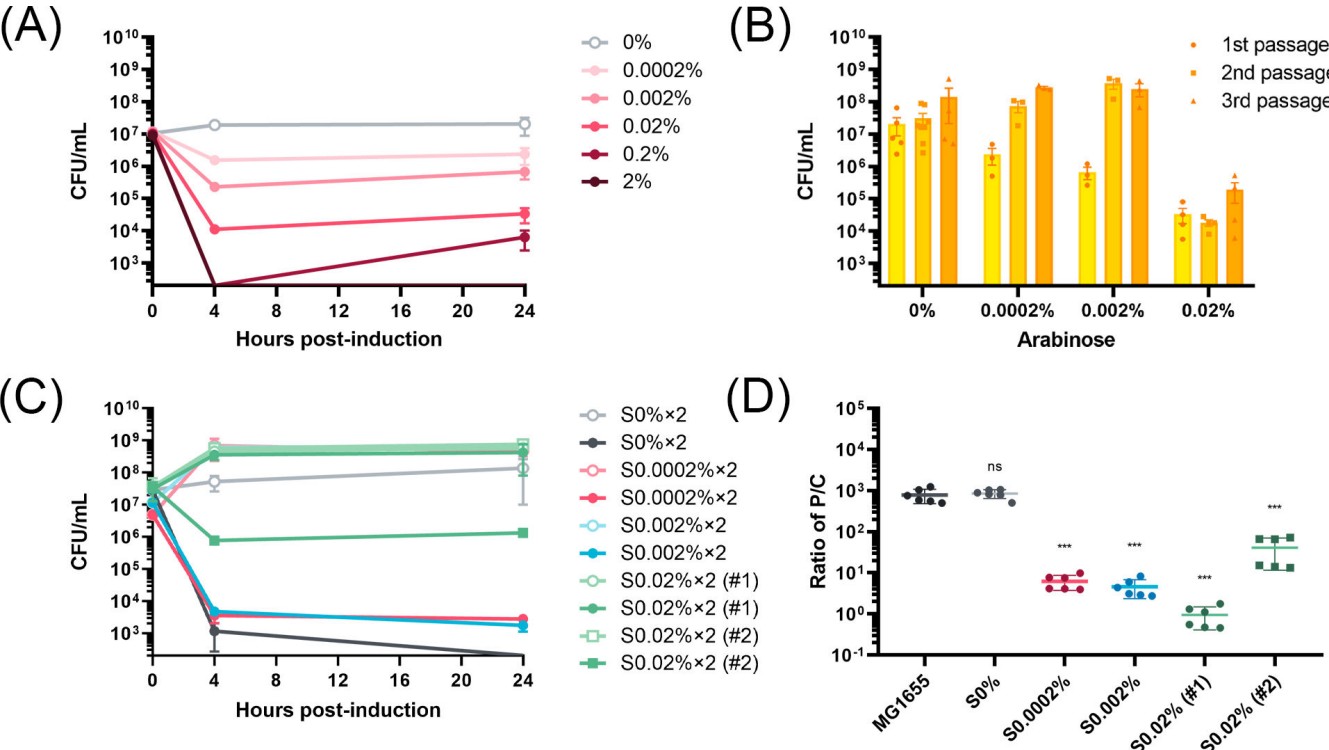

**FIG 5** Assessment of *E. coli* MG1655 cell viability and plasmid copy numbers upon induction of ParE2 protein expression. (A) The induction of ParE2 protein expression in *E. coli* MG1655 cells exhibited an induction-strength-dependent pattern, with 2% induction leading to a cell count below the detection limit (200 CFU/mL). (B) Cultures of MG1655 cells harboring ParE2-expressing plasmids were passaged every 24 h at the indicated induction levels, and all successive cultures had increased numbers of viable cells. (C) Cell viability was subsequently determined for two colonies isolated after two passages and re-induced with the addition of 0% (hollow circle) or 2% (solid circle) arabinose. All re-grown colonies apparently exhibited the same growth pattern at 0% reinduction, indicating there were no growth defects for those nonsensitive cells. Each data point represents the mean value of the two colonies, shown with SEM. (D) The PCNs of these surviving cells were determined by qPCR analysis. All cells that lost partial or complete sensitivity to reinduction exhibited a decreased PCN. The graph shows SD from two independent experiments performed in triplicates. The ratios of P/C of surviving cells were compared to that of the pre-induction MG1655 cells and an unpaired two-tailed Student's *t*-test was performed. ns, non-significant and ***$P < 0.001$.

strength. As before, following the second passage, two colonies from each induction strength were selected and re-grown, and ParE2 protein expression was reinduced with 0% or 2% arabinose (Fig. 5A). Strikingly, upon re-induction, there were varied toxicity profiles: one cell from the 0.02% induction group [S0.02% (#1)] had the survivor phenotype, colonies derived from the 0% induction group retained sensitivity to ParE2 expression induction, and all other cultures had an intermediate phenotype with partial survival (Fig. 5C).

Subsequent quantification of PCN showed a substantial reduction for all cultures exhibiting decreased sensitivity to ParE2 protein re-induction (Fig. 5D). Specifically, in the 0.0002% and 0.002% induction groups, the average reduction in PCN exceeded 125-fold, resulting in an average from 779.0 ± 300.0 for sensitive cells to 5.4 ± 2.3 plasmid copies per chromosome for insensitive cells. In the 0.02% induction group, one culture [S0.02% (#1)] with the survivor phenotype upon ParE2 protein re-induction exhibited an average 824-fold reduction in PCN (to 0.9 ± 0.5 plasmid copies per chromosome), while the other culture [S0.02% (#2)] that retained some sensitivity to ParE2 protein re-induction had an intermediate but significant 19-fold reduction in PCN (to 41.0 ± 29.5 plasmid copies per chromosome). Consistently, colonies derived from the 0% induction group, which retained sensitivity to ParE2 protein re-induction, exhibited no essential change in PCN.

## Reduced plasmid copy number is a generalized phenotype for *E. coli* survival of ParE toxin expression

We questioned if the reduction in PCN in response to ParE toxin expression was somehow limited to the MG1655 strain. To evaluate this, *E. coli* TOP10 cells were transformed with the pMindBAD plasmid encoding the ParE1 protein, allowing assessment of the viability and induction response in a different strain. We specifically selected the TOP10 strain, which lacks a functional *ara*BAD operon, including the *araB*, *araA*, and *araD* genes responsible for converting arabinose into metabolites that can be used by the cell (32). The selection of this strain allows a stable induction strength over the time course of the experiment. Induction of ParE1 protein in TOP10 *E. coli* cells demonstrated a direct correlation between the induction strength and the toxicity of ParE1 toxin (Fig. 6A). The trend is similar to that seen with the MG1655 strain (Fig. 1B), although this strain was much more sensitive to inducer and necessitated lowering the arabinose concentrations in subsequent experiments.

When the TOP10 strain cultures were subjected to the serial passage assay, they exhibited a pattern similar to that observed with ParE2 protein but not ParE1 protein in MG1655 cells: cultures exhibited a greater increase in CFU counts at low (0.00002%) and medium (0.0002%) induction strengths compared to high (0.002%) induction strengths (Fig. 6B). When colonies from each induction strength were isolated and re-grown, followed by re-induction of the ParE1 protein expression, all cells showed a decrease in CFU counts at the 4-h time point, followed by a robust recovery in growth, while cells from 0.02% induction group exhibited a complete cell loss within 24 h (Fig. 6C). Consistent with the MG1655 strain, only cultures with the ability to recover CFUs exhibited a reduction in PCN, with an average reduction exceeding 56-fold, resulting in an average plasmid copies per chromone change from 416.1 ± 53.2 to 7.4 ± 2.3 (Fig. 6D). This indicates that the reduction in PCN is independent of strain and instead arises from toxicity, or potentially from the specific mechanism of the toxins, rather than the specific cellular context.

Collectively, when considering both ParE toxin types and the two *E. coli* strains, a noteworthy trend emerges: out of the total 24 tested colonies, 20 colonies exhibited either partial or complete reduction in both toxicity profile and PCN, while in comparison, four colonies had no significant change in either toxicity profile or PCN. Continual exposure to the expressed toxin could enrich the population, with increased numbers of surviving cells in subsequent passages. ParE2 toxin protein exhibits a stronger toxicity profile than ParE1 toxin protein; hence, cells need to maintain a lower expression level of ParE2 to avoid being killed. These findings underscore the consistency of the observed

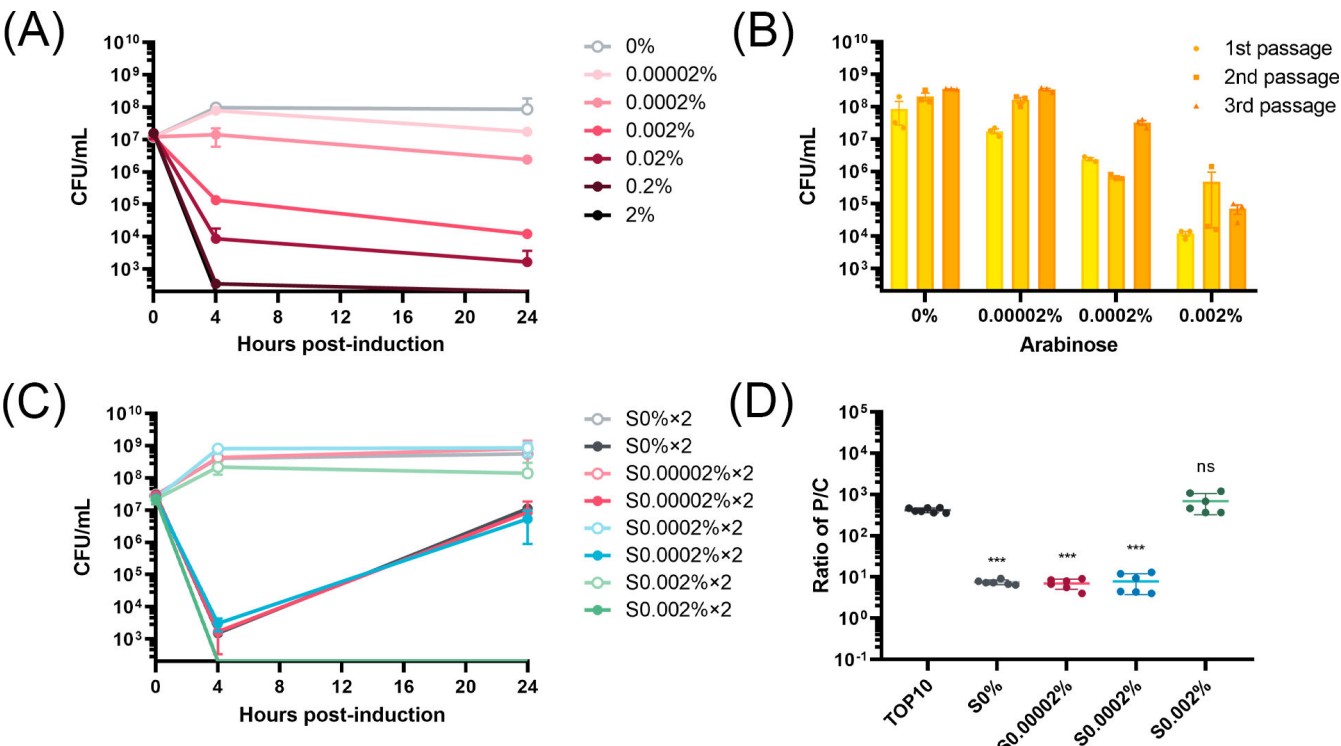

**FIG 6** Assessment of *E. coli* TOP10 cell viability and plasmid copy numbers upon induction of ParE1 protein expression. (A) The induction of ParE1 protein expression in *E. coli* TOP10 cells exhibited an induction-strength-dependent pattern. (B) Cultures of TOP10 cells harboring ParE1-expressing plasmids were passaged every 24 h at the indicated induction levels, and all successive cultures had increased numbers of viable cells. (C) Cell viability was subsequently determined for two colonies isolated after two passages and re-induced with the addition of 0% (hollow circle) or 2% (solid circle) arabinose. All re-grown colonies apparently exhibited the same growth pattern at 0% reinduction, indicating there were no growth defects for those nonsensitive cells. Each data point represents the mean value of the two colonies, shown with SEM. (D) The PCNs of these surviving cells were determined by qPCR analysis. All cells that lost partial or complete sensitivity to reinduction exhibited a decreased PCN. The graph shows SD from two independent experiments performed in triplicates. The ratios of P/C of surviving cells were compared to that of the pre-induction MG1655 cells and an unpaired two-tailed Student's *t*-test was performed. ns, non-significant and ***$P < 0.001$.

phenomenon across different conditions and reinforce the notion that survival by reducing PCN as a response to toxic protein expression is a reproducible and potentially adaptive cellular strategy.

## DISCUSSION

In this study, we observed a distinct phenotype characterized by a reduction in plasmid copy number following arabinose-induced expression of ParE proteins imparting levels of toxicity to *Escherichia coli* cells starting from the lag phase. This led to a robust (typically four logs) but not complete decrease in cell viability, followed by a consistent recovery of culture growth (Fig. 1B, 5A, and 6A). Furthermore, within the population of surviving cells, a subset exhibited a loss of apparent toxicity upon re-induction (Fig. 4C, 5C, and 6C). These findings collectively suggest that, while the initial induction of the ParE toxin expression leads to a reduction in cell viability, a portion of the bacterial population adapts to the toxic conditions and ultimately regains growth. Importantly, our examination of the plasmids carried by these insensitive cells revealed a consistent and stably reduced PCN (Fig. 4D, 5D, and 6D). This indicates that the reduction in PCN might serve as a reproducible mechanism enabling cells to mitigate the toxic effects of ParE toxin expression (Fig. 7).

Plasmid stability, which refers to the ability of a plasmid to persist within a population of host cells, has crucial impacts on the expression of genes encoded in the plasmid.

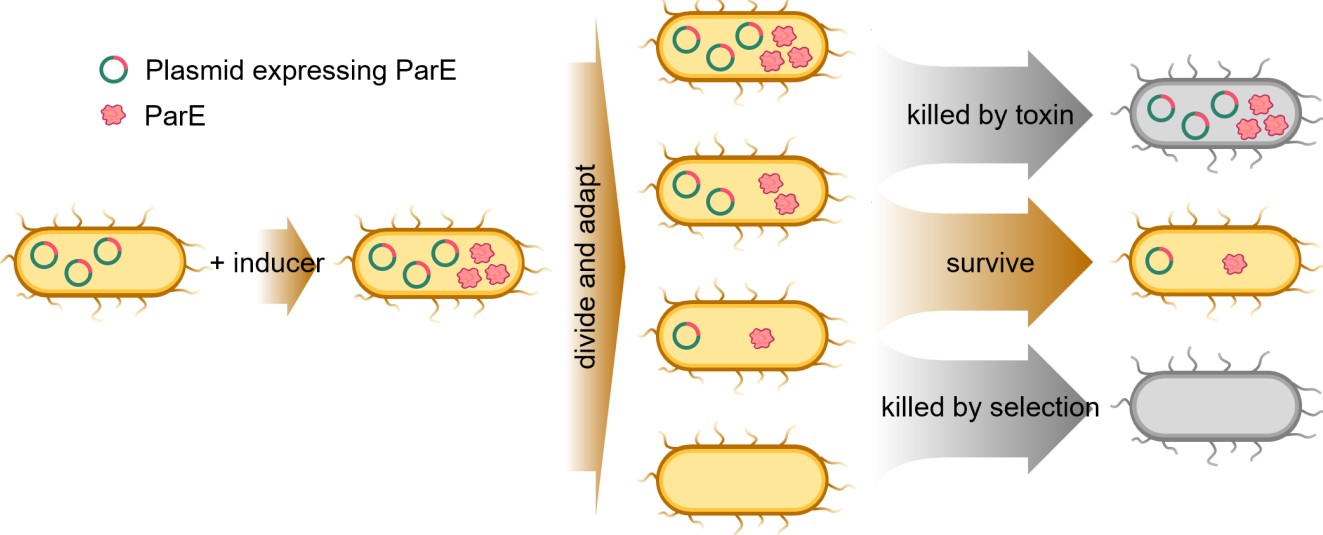

**FIG 7** Plasmid copy number reduction as a survival mechanism against ParE toxin proteins expressed in *E. coli*. In our study, the stationary cultures of *E. coli* cells harboring ParE-expressing plasmids were diluted in fresh media, and ParE toxin expression was induced by the addition of arabinose. Antibiotics were always added in the media to select for plasmid-harboring cells. When the plasmids were present in cells at high copy numbers, the cells were susceptible to the lethal dose of ParE toxin. When the plasmid copy number was reduced, the cells had a chance to survive exposure to a sub-lethal dose of ParE toxin. However, if the PCN was reduced beyond some threshold level, antibiotic selection would reduce cell survival due to the lower expression of the encoded antibiotic-resistant gene.

PCN is influenced by various factors, such as the origin of replication proteins, plasmid topology, segregation machinery, and selection pressure (33). Dumon-Seignovert et al. (34) observed that the overexpression of several heterologous proteins in *E. coli* cells becomes toxic and diminishes plasmid stability, as assessed by the percentage of colonies that could survive on both non-selection and selection agar plates in their study. In our investigation, we provide evidence that the toxic expression of ParE toxins leads to a reduction in PCN, directly impacting the accessible gene dosage for toxin expression. These findings raise important questions about potential confounding variables in studies that rely on the ectopic expression of genes to explore their physiological effects. Based on the current study, this phenotypic adaptation arises within 4–8 h of induction. In the current study, this effect correlates directly with the extent of toxicity, such that more toxic expression results in quicker adaptation. The observed reduction in PCN suggests that gene expression studies conducted in plasmid-based systems may not accurately reflect the physiological responses of cells, particularly when dealing with toxic proteins.

Plasmids used in this study are ColE1-like plasmids featuring a pUC origin, the replication and maintenance of which rely on the dynamic balance between two small RNAs, RNA I and RNA II. Our findings indicated a notable decrease in the transcription level of RNA II within these insensitive cells with reduced PCN (Fig. S8). Notably, the transcription level of RNA I was undetectable under these conditions. This observation suggests that the decrease in PCN might be accompanied by a proportional reduction in RNA I's absolute transcript level, potentially leading to its undetectability. Further investigations into the expression of RNA I in these insensitive cells to reveal the balance change between these two RNAs would provide insight into the molecular regulation of the plasmid.

On the other hand, our analysis, comparing the complete genomes of *E. coli* cell samples S002 and S2 (Fig. 1) with the *E. coli* K-12 MG1655 reference sequences (sourced from GenBank accessions U00096.1, U00096.2, and U00096.3, and ATCC references 700926 and 47076), revealed seven variants in the reference genomes and our samples as detailed in Table S3. Among these variants, only the N845S substitution in the DNA

polymerase I (polA) appeared exclusively in our samples. This mutation may be causally linked to the observed phenotype of reduced PCN. Research has highlighted the crucial role that DNA polymerase I plays in the replication of ColE1-like plasmids within *E. coli*. It has been observed that mutants deficient in DNA polymerase I activity are incapable of sustaining the ColE1 plasmid (35, 36). Furthermore, the N845 residue has been reported to be involved in the recognition of correct terminal base pairs in the DNA substrate (37) and in the incorporation of ribonucleotides (38) and contributes to the polymerase fidelity (39) and catalytic activity (40). The surviving S002 and S2 cells both demonstrated the same mutation in DNA polymerase I, a change we hypothesize arose as an adaptive response to evade the effects of ParE toxin expression. This mutation likely resulted from the toxin's gyrase inhibition, representing a random yet prevalent adaptation selected in our study to mitigate toxicity while promoting survival. To explore this hypothesis further, an investigation into the N845S mutation's impact on DNA replication and expression of other gyrase-inhibiting agents would offer deeper insights into the mechanisms driving the observed reduction in PCN.

In conclusion, our studies highlight a potential technical limitation to studies of toxic proteins in engineered inducible systems, as well as the phenotypic balance between two opposing effects of the ParE toxin. The ability of some cells to recover or become insensitive to toxin expression when exposed to arabinose again raises intriguing questions about the underlying mechanisms of this phenomenon. It also underscores the complexity of bacterial responses to toxic proteins and the role of plasmid copy number in mediating these responses. Further investigation is needed to elucidate the precise mechanisms involved in PCN determination and to apply these findings in various biotechnological and medical contexts.

## MATERIALS AND METHODS

### Bacterial strains and plasmids

The bacterial strains, plasmids, and primers used in this study are listed in Tables S1 and S2. Standard protocols were followed for DNA manipulation, PCR cloning, and DNA purification. DNA fragments were PCR amplified from *Mycobacterium tuberculosis* strain H37Rv genomic DNA and plasmid DNA using Q5 High-Fidelity DNA Polymerase (NEB) and assembled by Gibson Assembly using Gibson Assembly Master Mix (NEB) according to the manufacturer's instructions. The chemically competent cells of *Escherichia coli* MG1655 or TOP10 strains were prepared according to standard protocols and aliquoted as "starting" cells for transformation. Plasmid was transformed into competent cells by heat shock method according to standard protocols. Plasmid was extracted and purified using Zyppy Plasmid Miniprep Kit (Zymo Research) and sequenced by GENE-WIZ (USA) or Plasmidsaurus (USA). Genomic DNA was extracted and purified using Wizard Genomic DNA Purification Kit (Promega) and sequenced by Plasmidsaurus (USA). Genome annotations are sourced from GenBank: U00096.3.

### Construction of plasmids

#### pMindBAD::mtparE1

The fragments of the *mtparE1* gene and the pMind vector were separately PCR amplified and subsequently assembled to generate the pMind::*mtparE1* plasmid. This construction process involved utilizing the mtparE1 primer set for the gene fragments and the pMind primer set for the vector. The *tetRO* promoter of pMind::*mtparE1* was substituted with the *araC*-pBAD promoter to generate the pMindBAD::*mtparE1* plasmid. This replacement involved assembling the fragment of the *araC*-pBAD promoter, which was amplified from pHerd20T using the araC-pBAD primer set, with the fragment amplified from pMind::*mtparE1* using the pMind + araC pBAD primer set.

*pMindBAD::Strep-mtparE2*

The fragments of the *mtparD2E2* gene and the pMind vector were separately PCR amplified. These fragments were then assembled to generate the pMind::*mtparD2E2* plasmid, using mtparD2E2 and pMind primer sets. The *mtparD2* gene of pMind::*mtparD2E2* was subsequently removed, and a *Strep* tag was fused upstream of the *mtparE2* gene to generate the pMind::*Strep-mtparE2* plasmid. This modification was achieved by performing a PCR amplification of pMind::*mtparD2E2* using the mtparE2 + Strep primer set. The resulting amplicon was subjected to KLD treatment using the KLD Enzyme Mix (NEB, Cat# M0554S), according to the manufacturer's instructions. Finally, the *tetRO* promoter was replaced with the *araC*-pBAD promoter to generate the pMindBAD::*Strep-mtparE2* plasmid, following the same method as described above.

## Growth of cultures and measurements of cell viability

Overnight cultures were inoculated from −80°C frozen 20% glycerol stocks or single colonies on LB agar plates into LB media supplemented with appropriate antibiotic and 1% glucose and grown overnight (18–20 h) at 37°C with shaking at 200 rpm. The overnight cultures were then back-diluted 1:20 in fresh LB media and added with arabinose to final concentrations of 0%–2% and grown at 37°C with shaking at 200 rpm. Aliquots of cultures were collected at the defined time intervals, and 10-fold serial dilutions in sterile 0.9% saline solution were performed and spotted onto LB agar plates supplemented with 50 µg/mL kanamycin and 1% glucose for the determination of the colony-forming units. The CFUs were subsequently inverted to CFU/mL based on the initial spotting volumes. Kanamycin was added at 50 µg/mL. Carbenicillin was added at 100 µg/mL. Gentamycin was added at 10 µg/mL. The limit of detection for CFU/mL measurements was 200 CFU/mL. Figures were generated using GraphPad Prism 8.01.

## Determination of the ratio of plasmid to chromosome per cell

### DNA sample preparation for qPCR

The method described previously by Artarini et al. (41, 42) was modified in this study. Briefly, overnight cultures were inoculated from frozen glycerol stocks into 8 mL of LB media supplemented with 50 µg/mL kanamycin and grown overnight (18–20 h) at 37°C with shaking at 200 rpm. The cells were then harvested through centrifugation and resuspended in 80 µL of distilled water. The cell suspension was heated at 99°C for 20 minutes, frozen at −80°C, and heated again at 99°C for 20 minutes. Supernatants were collected by centrifugation at 16,000 × *g* for 10 minutes, followed by a series of 10-fold dilutions as a template source.

### Design of primer sets for qPCR and RT-qPCR

The constitutively expressed *gapA* gene, encoding D-glyceraldehyde-3-phosphate dehydrogenase, was chosen as the reference gene to normalize the expression levels of target genes. The primer sequences for *gapA* were sourced from the study by Robbins-Manke et al. (43). Primers for the target genes were designed with a similar melting temperature to the gapA primer set using IDT PrimerQuest Tool. Primers were synthesized by IDT (USA). Primer specificity was tested by the PCR product electrophoretogram and the product melting curve analysis following qPCR amplification.

### Real-time qPCR using SYBR Green dye

Four microliters of at least three template dilutions (e.g., $10^{-1}$, $10^{-2}$, and $10^{-3}$ dilutions) was then amplified using Fast SYBR Green Master Mix (Applied Biosystems) and the appropriate primer set in separated reaction on Roche LightCycler 480 II Real-Time PCR system using the following cycling conditions for all amplicons: 5 minutes at 95°C (pre-incubation), followed by 45 cycles of 5 s at 95°C, 30 s at 58°C, and 20 s at 58°C. At the end, a dissociation stage was added: 5 s at 95°C and 1 minute from 60°C to 97°C. Cycle

threshold (Ct) values were determined automatically using the built-in analysis function of "Abs Quant/Fit Points" in LightCycler 480 software version 1.5.1.62.

The determination of the ratio of the plasmid to chromosome (P/C) was reported by Artarini et al. (41) and Skulj et al. (42). Briefly, in each run, the curve for each primer set was constructed by placing the log value of the dilution fold (e.g., −1, −2, and −3) on the x axis and Ct value on the y axis. The calculation of amplification efficiency (E) involved utilizing the average slope value from the curves obtained with the same primer set in the same run, as per the formula outlined in equation (1).

$$E = 10^{-\frac{1}{\text{slope}}} \tag{1}$$

The determination of the ratio of target A to target B (A/B) was accomplished by applying equation (2), which considered distinct amplification efficiencies (E) and Ct values corresponding to the targets.

$$A/B = \frac{E_A^{-Ct_A}}{E_B^{-Ct_B}} \tag{2}$$

The ratio of P/C was defined as the ratio of plasmid (parE gene, target A) to chromosome (gapA gene, target B) and determined across all dilutions within each sample, followed by the calculation of the mean and standard deviation (SD) for these values. Figures were generated using GraphPad Prism 8.01.

## Determination of the ratio of RNA II to RNA I

### cDNA sample preparation for RT-qPCR

Overnight cultures were inoculated from frozen glycerol stocks into LB media supplemented with appropriate antibiotics and 1% glucose and grown overnight (18–20 h) at 37°C with shaking at 200 rpm. The overnight cultures were then back-diluted 1:20 in 50 mL of fresh LB media supplemented with appropriate antibiotic and grown at 37°C with shaking at 200 rpm. The cells were then harvested through centrifugation when an optical density (OD$_{600}$) of 0.6–0.8 was reached. Total RNA was extracted using the Direct-zol RNA Miniprep Plus Kit (Zymo Research, Cat# R2072) according to the manufacturer's instructions. Contaminating genomic and plasmid DNA was removed, and total RNA was cleaned using RNA Clean & Concentrator Kit (Zymo Research, Cat# R1013) according to the manufacturer's instructions. DNA contamination was tested by qPCR using appropriate primer sets. RNA samples with a Ct value of not less than 35 were considered to contain limited or no DNA contamination. Once the RNA sample was free of DNA contamination, it was reverse-transcribed into cDNA using the High-Capacity cDNA Reverse Transcription Kits (Applied Biosystems, Cat# 4368814), followed by a series of 10-fold dilutions as a template source. When the elimination of DNA contamination cannot be accomplished, a control group was introduced, i.e., the reverse transcriptase was replaced with an equal volume of water in the reverse transcription reaction.

### Real-time qPCR using SYBR Green dye

Four microliters of cDNA template or control template was then amplified by qPCR as described above. The ratio of RNA II (target A) to RNA I (target B) was determined across all dilutions within each sample as described above by applying equation (2) or (3), followed by the calculation of the mean and SD for these values. Figures were generated using GraphPad Prism 8.01.

$$A/B = \frac{\left(E_A^{-Ct_A}\right)_{exp} - \left(E_A^{-Ct_A}\right)_{Ctrl}}{\left(E_B^{-Ct_B}\right)_{exp} - \left(E_B^{-Ct_B}\right)_{Ctrl}} \qquad (3)$$

## Determination of the *araE* mRNA

Overnight cultures were inoculated from frozen glycerol stocks into LB media supplemented with appropriate antibiotic(s) and grown overnight (15 h) at 37°C with shaking at 200 rpm. The overnight cultures were divided into two portions. Arabinose was added to one portion to achieve a final concentration of 2%, and the same amount of water was added to the other portion as a control (0%). Subsequently, both portions were grown at 37°C with shaking at 200 rpm for 5 h. cDNA samples were prepared, and data analysis was performed as described above.

## Imaging of fluorescent cells

Overnight cultures were inoculated from frozen glycerol stocks into LB media supplemented with appropriate antibiotic(s) and 1% glucose and grown overnight (15 h) at 37°C with shaking at 200 rpm. Arabinose was then added to a final concentration of 0.2% or 2% to induce mCherry protein expression and grown at 37°C with shaking at 200 rpm for 4–5 h. Five microliters of the cultures was applied to a microscope slide (Fisher Scientific, USA), and a cover glass (Fisher Scientific, USA) was then sealed into place. The cells were imaged using a Leica Microsystems Model DMi8 with appropriate settings.

## ACKNOWLEDGMENTS

This research is funded by the Department of Defense under grant number W81XWH-20-1-0121. Financial support for publication was provided by the University of Oklahoma Libraries' Open Access Fund.

The author would like to thank Bourne lab members for their suggestions and assistance with various experiments.

## AUTHOR AFFILIATION

[1]Department of Chemistry and Biochemistry, University of Oklahoma, Norman, Oklahoma, USA

## AUTHOR ORCIDs

Shengfeng Ruan ⓘ http://orcid.org/0000-0002-0218-8575
Christina R. Bourne ⓘ http://orcid.org/0000-0001-6192-3392

## FUNDING

| Funder | Grant(s) | Author(s) |
| --- | --- | --- |
| U.S. Department of Defense (DOD) | W81XWH-20-1-0121 | Christina R. Bourne |

## ADDITIONAL FILES

The following material is available online.

### Supplemental Material

**Supplemental material (Spectrum03973-23-s0001.pdf).** Tables S1-S3; Fig. S1-S8.

### Open Peer Review

**PEER REVIEW HISTORY (review-history.pdf).** An accounting of the reviewer comments and feedback.

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
