## [Reviewer comments · Microbiology Spectrum]

Microbiology Spectrum

***Escherichia coli* Cells Evade Inducible ParE Toxin Expression by Reducing Plasmid Copy Number**

Shengfeng Ruan and Christina Bourne

Corresponding Author(s): Christina Bourne, The University of Oklahoma

Review Timeline:

Submission Date:	November 17, 2023
Editorial Decision:	February 1, 2024
Revision Received:	March 29, 2024
Accepted:	April 15, 2024

Editor: Cezar Khursigara

Reviewer(s): The reviewers have opted to remain anonymous.

Transaction Report:

DOI: <https://doi.org/10.1128/spectrum.03973-23>

Re: Spectrum03973-23 (*Escherichia coli* Cells Evade Inducible ParE Toxin Expression by Reducing Plasmid Copy Number)

Dear Dr. Christina R Bourne:

Thank you for the privilege of reviewing your work. Below you will find my comments, instructions from the Spectrum editorial office, and the reviewer comments.

Though both reviewers found the study interesting and worthy of publishing, they point out several major shortcoming. Addressing those would definitely strengthen your conclusions and improve readability of the paper.

Revision Guidelines

Sincerely,
Elitza Tocheva
Editor
Microbiology Spectrum

Reviewer #1 (Comments for the Author):

The authors describe experiments to explain how some *E. coli* MG1655, when overexpressing *parE* (the toxin component of a toxin/antitoxin pair), become tolerant of the toxin. Using an elegant and logical approach, they show this trait is inheritable, chromosomally located, and not driven through increased uptake of arabinose (*parE* is heterologously expressed using the

araBAD promoter). They also demonstrate this phenotype isn't strain dependent as it can be recapitulated in *E. coli* TOP10, and that resistance develops in an analogous manner when overexpressing a different DNA gyrase inhibitor called ParE2. Their data supports a model where chromosomal mutations lead to a decreased plasmid copy number, which results in less toxin production. These data serve as a reminder that asking bacteria to express high levels of plasmid-encoded proteins may select for populations with reduced plasmid copy number.

Broader comments; none of these need to be experimentally addressed but I feel the authors should at least address:

1. Have the authors performed any experiments to attempt to identify the nature of the mutations on the chromosome that decrease plasmid copy number? The authors make a few comments along the way suggesting so (for example, RNAI and II detection in the Discussion). The obvious question any reader will have, is whether the authors whole genome sequenced any mutants? If so, were there any clues or just a mess of SNPs/deletions that didn't provide any insight?
2. Chromosomal mutations could be (at least): a. those that decrease toxin impact on DNA gyrase, which happens to impact plasmid copy number, b. those that directly impact copy number of a narrow set of replicons, c. those that impact plasmid copy number regardless of incompatibility group. Do the authors have any insight(s) favoring one model over the other?
3. Do the mutations affect fitness in any measurable way, for example growth in minimal medium?

Specific comments:

Line 75: do you mean the toxin isn't active in *E. coli*? If so, maybe it would clarify this sentence if you said "...noted toxins that are active in X, Y, and Z, but not in *E. coli*"

Lines 83, 85, others: Personal preference, I feel it sounds awkward saying "they found...". I'd suggest (but not require) either "That study reported...", "Gupta et al found...", or something similar

Lines 129-130: the text says Fig 1D = S02 and Fig 1E = S002, but the labeling in the Figure is the reverse.

The Figure legend of Fig 1G is unclear; I assume these are induced and uninduced? On a related note, Figures should be understandable from descriptions provided in the Figure legends, without needing to refer to the main text. Please make sure all Figure legends have sufficient descriptions.

Line 160: I'm not sure this conclusion is accurate, as you don't directly measured protein expression.

Line 175: by "starting", do you mean "wild-type"?

Fig S4: Its unclear to me, why panel A does not include S002 and S2 constructs, and why panel B does not include the MG1655 construct?

Lines 220-221: It does not appear from Fig 4B, that the 2% induction becomes "progressively higher", as the 3rd passage appears less resistant than the 2nd.

Fig 5A: I assume the 0.2% and 2% line overlap? If so, suggest mentioning this to the reader in the Figure legend

Line 268-269: It appears the 3rd passage may have higher numbers than the other two? Do the authors have stats to confirm there is no sig difference between the three measurements from 0.02% ara?

Reviewer #2 (Comments for the Author):

In this manuscript, Ruan and Bourne investigate the mechanism underlying resistance to parE toxicity in strains that have experienced ParE overexpression. The authors conclude, based on a qPCR assay, that *Mtb* parE1 overexpression results in reduced plasmid copy number. It is concluded that the reduced number of plasmid copies reduces toxicity of induced parE expression.

Some questions and comments for the authors to consider are listed below:

A lot of interpretation hinges on the claim that the plasmids contain no mutations. More detail on how this was determined, including how the sequence data were assembled, would be helpful for readers. The authors should perhaps upload their raw sequence data.

The authors seem to be basing their conclusion that parE expression levels are lower in the induction resistant strains solely on CFU counts. Are there experimental data where they have assessed ParE protein or parE transcript?

The PCR assay to measure copy number is reasonable, but the conclusions would be better supported if (only in one

background) the authors had some evidence of lower copy number using a second approach, e.g. constitutive expression of a fluorescent protein from their plasmid in a high copy and low copy background. This could stem concerns about possible systematic differences in PCR amplification between samples that could arise due to PCR inhibiting compounds that might exist in the resistant ("low copy") strains.

Do the induction resistant strains grow slower than wild type? One could imagine models whereby slow growth mitigates the toxic effects of parE expression.

Figure 2 legend needs some information on the plasmid and origin of the fluorescent signal.

Whole genome sequence data would be very welcome in this study, and could provide better understanding of what's going on with these results. The E. coli genome could be sequenced, assembled, and run through breseq for approximately \$100 at seqcenter or seqcoast.

Questions about clarity of language usage and writing.

Line 57: "Notably, arabinose is usually found at significantly lower concentrations...." This sentence is unclear. Where are concentration of arabinose lower?

Lines 83 and 85: "they found..." who is they? Please revise.

Line 94-95: "we find that the loss of reintroduction does not result..." The use of "loss of reintroduction" throughout doesn't really make sense to me. Is it easier to simply call this phenomenon "ParE toxicity resistance"?

Reviewer #1 (Comments for the Author):

We thank the reviewers for their careful analysis of our work. We have incorporated the suggestions, including additional experiments, that provide more conclusive mechanism(s) for the observed phenotype. We have also made many small changes to the text to improve clarity throughout.

The authors describe experiments to explain how some *E. coli* MG1655, when overexpressing parE (the toxin component of a toxin/antitoxin pair), become tolerant of the toxin. Using an elegant and logical approach, they show this trait is inheritable, chromosomally located, and not driven through increased uptake of arabinose (*parE* is heterologously expressed using the araBAD promoter). They also demonstrate this phenotype isn't strain dependent as it can be recapitulated in *E. coli* TOP10, and that resistance develops in an analogous manner when overexpressing a different DNA gyrase inhibitor called ParE2. Their data supports a model where chromosomal mutations lead to a decreased plasmid copy number, which results in less toxin production. These data serve as a reminder that asking bacteria to express high levels of plasmid-encoded proteins may select for populations with reduced plasmid copy number.

Broader comments; none of these need to be experimentally addressed but I feel the authors should at least address:

1. Have the authors performed any experiments to attempt to identify the nature of the mutations on the chromosome that decrease plasmid copy number? The authors make a few comments along the way suggesting so (for example, RNAi and II detection in the Discussion). The obvious question any reader will have, is whether the authors whole genome sequenced any mutants? If so, were there any clues or just a mess of SNPs/deletions that didn't provide any insight?

Response: Thank you for this suggestion, and we agree this is of high interest. As such, we carried out whole genome sequencing for cell samples S002 and S2, which are insensitive to induction after ParE exposure. When compared to several reference *E. coli* K-12 MG1655 genome sequences, our analysis identified a N845S substitution in the DNA polymerase I (*polA*). This asparagine residue at position 845 has been demonstrated in the literature to be critical for the polymerase's activity and fidelity. Importantly, it has been observed that mutants deficient in DNA polymerase I activity are incapable of sustaining the ColE1-like plasmid. We speculate that this is likely the mechanism by which cells reduced PCN.

2. Chromosomal mutations could be (at least): a. those that decrease toxin impact on DNA gyrase, which happens to impact plasmid copy number, b. those that directly impact copy number of a narrow set of replicons, c. those that impact plasmid copy number regardless of incompatibility group. Do the authors have any insight(s) favoring one model over the other?

Response: Based on our whole genome and plasmid sequencing results, no mutations were detected in the DNA gyrase genes or within the plasmid itself. As discussed above, we speculate that the N845S substitution identified in the DNA polymerase I (*polA*) may be the underlying genetic change responsible for the observed reduction in PCN in our cell samples.

3. Do the mutations affect fitness in any measurable way, for example growth in minimal medium?

Response: The observed mutations did not result in any noticeable alteration to the growth patterns under the conditions tested (for example, Fig. 4C demonstrates the same viability cell counts for both induction “insensitive” and sensitive cells, suggesting growth is not impacted).

Specific comments:

Line 75: do you mean the toxin isn't active in *E. coli*? If so, maybe it would clarify this sentence if you said "...noted toxins that are active in X, Y, and Z, but not in *E. coli*"

Response: We have clarified this: (line 73 now) “We have previously noted toxins that are active in their native hosts but were not toxic in the commonly used *Escherichia coli* surrogate host (22), and in the current work have uncovered additional confounding issues with these strategies.”

Lines 83, 85, others: Personal preference, I feel it sounds awkward saying "they found...". I'd suggest (but not require) either "That study reported...", "Gupta et al found...", or something similar

Response: We have changed them as follows:

Lines 83: (line 81 now) “However, it was reported that under some growth conditions the plasmid could be lost, and viability restored over time.”

Line 85: “It was found that the overexpression in an *E. coli* host caused potent toxicity but was followed by a full recovery of growth.”

Lines 129-130: the text says Fig 1D = S02 and Fig 1E = S002, but the labeling in the Figure is the reverse.

Response: Our apologies! Yes, they are reversed. It is changed to (lines 131-133) “... as did the culture originating from the 0.2% induced surviving colony (S02, Fig. 1E). However, there was no sensitivity remaining for cells originating from 0.02% induction (S002, Fig. 1D) and 2% induction (S2, Fig. 1F) conditions.”

The Figure legend of Fig 1G is unclear; I assume these are induced and uninduced? On a related note, Figures should be understandable from descriptions provided in the Figure legends, without needing to refer to the main text. Please make sure all Figure legends have sufficient descriptions.

Response: The legend is changed to: (G) The induction of ParE1 protein expression from the extracted plasmids exhibited a strong toxicity profile to the “starting” *E. coli* MG1655 cells.

Line 160: I'm not sure this conclusion is accurate, as you don't directly measured protein expression.

Response: Yes, we have altered the wording to indicate that we are not directly referring to measure protein levels. We have numerous attempts to capture expression of ParE by Western blots, but given the toxic nature of ParE to cells that limits their growth, these have never produced convincing signals. Our inference is that ParE toxin is expressed based on viability assays rather than direct protein

expression analyses. Further, it seems likely that very few ParE toxins are required to have a dramatic impact on viability. This is an area of active and on-going interest in our group.

Line 175: by "starting", do you mean "wild-type"?

Response: Yes.

Fig S4: Its unclear to me, why panel A does not include S002 and S2 constructs, and why panel B does not include the MG1655 construct?

Response: Fig. S4A focuses on evaluating the constitutive expression of the AraE protein in *E. coli*, illustrating that AraE protein is well-expressed even in the absence of arabinose. This setup was specifically chosen to highlight AraE expression patterns without the complexity introduced by the S002 and S2 constructs. As for Fig. S4B, the inclusion of the MG1655 construct is deliberately omitted here because it is already analyzed in Fig. 2A, where its behavior in the context of the study is fully addressed.

Lines 220-221: It does not appear from Fig 4B, that the 2% induction becomes "progressively higher", as the 3rd passage appears less resistant than the 2nd.

Response: We agree that this trend appears evident; however it is not a significant difference (p -value of 0.17).

Fig 5A: I assume the 0.2% and 2% line overlap? If so, suggest mentioning this to the reader in the Figure legend

Response: They do not overlap. After 4 hr, 2% has a CFU/mL under detection limit. The legend is changed to: (A) The induction of ParE2 protein expression in *E. coli* MG1655 cells exhibited an induction-strength-dependent pattern, with 2% induction leading to a cell count below the detection limit (200 CFU/mL).

Line 268-269: It appears the 3rd passage may have higher numbers than the other two? Do the authors have stats to confirm there is no sig difference between the three measurements from 0.02% ara?

Response: No significant difference was found among these three passages at 0.02%.

Reviewer #2 (Comments for the Author):

We thank the reviewers for their careful analysis of our work. We have incorporated the suggestions, including additional experiments, that provide more conclusive mechanism(s) for the observed phenotype. We have also made many small changes to the text to improve clarity throughout.

In this manuscript, Ruan and Bourne investigate the mechanism underlying resistance to parE toxicity in strains that have experienced ParE overexpression. The authors conclude, based on a qPCR assay, that Mtb parE1 overexpression results in reduced plasmid copy number. It is concluded that the reduced number of plasmid copies reduces toxicity of induced parE expression.

Some questions and comments for the authors to consider are listed below:

A lot of interpretation hinges on the claim that the plasmids contain no mutations. More detail on how this was determined, including how the sequence data were assembled, would be helpful for readers. The authors should perhaps upload their raw sequence data.

Response: The resulting sequences are detailed which verified the absence of mutations in the plasmids, please refer to Fig. S2. These sequences were obtained using Nanopore technology (Plasmidsaurus), which provides a readout of the entire sequence (no assembly required); we have added the times coverage to the results section to support this method. The sequencing results of the plasmids show them to be nearly identical to the reference sequence, with only a minor number of mismatches occurring in backbone regions whose functions remain undefined. These findings underpin our assertion that plasmids are essentially mutation-free.

The authors seem to be basing their conclusion that parE expression levels are lower in the induction resistant strains solely on CFU counts. Are there experimental data where they have assessed ParE protein or parE transcript?

Response: Given the toxic nature of ParE to cells, it seems very few ParE molecules have a dramatic impact on viability, making direct measurement of its expression challenging. Our inference is that ParE toxin is expressed based on viability assays rather than direct protein expression analyses. We didn't assess the *parE* transcript, but we have assessed the RNA II transcript, which showed a reduced level, consistent with the reduced PCN. Hence we assume the *parE* transcript will be reduced consistently.

The PCR assay to measure copy number is reasonable, but the conclusions would be better supported if (only in one background) the authors had some evidence of lower copy number using a second approach, e.g. constitutive expression of a fluorescent protein from their plasmid in a high copy and low copy background. This could stem concerns about possible systematic differences in PCR amplification between samples that could arise due to PCR inhibiting compounds that might exist in the resistant ("low copy") strains.

Response: We agree and this is why we pursued expression of the mCherry protein, as discussed in Fig. 2.

Do the induction resistant strains grow slower than wild type? One could imagine models whereby slow growth mitigates the toxic effects of parE expression.

Response: The observed mutations did not result in any noticeable alteration to the growth patterns under the conditions tested (for example, Fig. 4C demonstrates the same viability cell counts for both induction “insensitive” and sensitive cells, suggesting growth is not impacted).

Figure 2 legend needs some information on the plasmid and origin of the fluorescent signal.

Response: Thank you. The legend has been changed to clarify this: (A) Fluorescence microscopy of the “starting” MG1655 as well as the S002 and S2 cells with ParE-expressing plasmid removed (“cured”) induced for the expression of fluorescent protein mCherry from a plasmid, pHerd20T, also under control of the arabinose inducible promoter. However, fluorescence of mCherry protein is only visualized in the “starting” MG1655 cells.

Whole genome sequence data would be very welcome in this study, and could provide better understanding of what's going on with these results. The *E. coli* genome could be sequenced, assembled, and run through breseq for approximately \$100 at seqcenter or seqcoast.

Response: Yes, we agree this is of high interest and chose to pursue this for the revised version. We have now included results for whole genome sequencing of cell samples S002 and S2. When compared to several reference *E. coli* K-12 MG1655 genome sequences, our analysis identified a N845S substitution in the DNA polymerase I (polA). This asparagine residue at position 845 has been demonstrated in the literature to be critical for the polymerase's activity and fidelity. Importantly, it has been observed that mutants deficient in DNA polymerase I activity are incapable of sustaining the ColE1-like plasmid. We speculate that this is likely the mechanism by which cells reduced PCN.

Questions about clarity of language usage and writing.

Line 57: "Notably, arabinose is usually found at significantly lower concentrations...." This sentence is unclear. Where are concentration of arabinose lower?

Response: Sentence is changed to (line 58 now) “Notably, due to catabolite repression, the induction of araBAD is easily repressed by the presence of excess glucose or lactose (11).”

Lines 83 and 85: "they found..." who is they? Please revise.

Response: Line 83: Roberts et al. Line 85: Gupta et al.

Sentences are changed to as follows:

Lines 83: (line 81 now) “However, it was reported that under some growth conditions the plasmid could be lost, and viability restored over time.”

Line 85: (line 84 now) “It was found that the overexpression in an *E. coli* host caused potent toxicity but was followed by a full recovery of growth.”

Line 94-95: "we find that the loss of reintroduction does not result..." The use of "loss of reintroduction"

throughout doesn't really make sense to me. Is it easier to simply call this phenomenon "ParE toxicity resistance"?

Response: Sentence is changed to: (lines 93-94 now) "Through culture-based assays, we find that the loss of toxicity upon re-induction of ParE protein expression does not result from mutations of the plasmids or by alteration of arabinose uptake."

Re: Spectrum03973-23R1 (*Escherichia coli* Cells Evade Inducible ParE Toxin Expression by Reducing Plasmid Copy Number)

Dear Dr. Christina R Bourne:

Your manuscript has been accepted, and I am forwarding it to the ASM production staff for publication. Your paper will first be checked to make sure all elements meet the technical requirements. ASM staff will contact you if anything needs to be revised before copyediting and production can begin. Otherwise, you will be notified when your proofs are ready to be viewed.

Sincerely,
Cezar Khursigara
Editor
Microbiology Spectrum